# Fossil Biomarkers and Biosignatures Preserved in Coprolites Reveal Carnivorous Diets in the Carboniferous Mazon Creek Ecosystem

**DOI:** 10.3390/biology11091289

**Published:** 2022-08-30

**Authors:** Madison Tripp, Jasmina Wiemann, Jochen Brocks, Paul Mayer, Lorenz Schwark, Kliti Grice

**Affiliations:** 1Western Australian Organic and Isotope Geochemistry Centre, The Institute for Geoscience Research, School of Earth and Planetary Sciences, Curtin University, Kent Street, Bentley, WA 6102, Australia; 2Department of Earth & Planetary Sciences, Yale University, 210 Whitney Avenue, New Haven, CT 06511, USA; 3Division of Geological and Planetary Sciences, California Institute of Technology, 1200 E. California Blvd., Pasadena, CA 91125, USA; 4Research School of Earth Sciences, The Australian National University, Canberra, ACT 2601, Australia; 5The Field Museum, 1400 S Lake Shore Dr., Chicago, IL 60605, USA; 6Organic Geochemistry Unit, Institute of Geoscience, Christian-Albrechts-University, 24118 Kiel, Germany

**Keywords:** steroids, diet, coprolites

## Abstract

**Simple Summary:**

Coprolites (fossilised faeces) can preserve important dietary information through geological time, offering insights into extinct animal diets. When digestion of dietary items leaves no unambiguous morphology to reconstruct the food spectrum of a coprolite producer, preserved biomolecular information can offer unique perspectives into the individual dietary composition and trophic relationships in ancient ecosystems. In this study we combine a uniquely diverse array of chemical techniques to demonstrate that biomarkers and macromolecular biosignatures from Carboniferous coprolites can reveal the dietary spectrum and trophic position of their extinct producers: an overwhelming abundance of cholesteroids, biomarkers of animal cholesterol, and an animal-affinity of the preserved macromolecular phase revealed by the statistical analysis of *in situ* Raman spectra, indicate a likely carnivorous diet for the coprolite producer. The presence of intact primary metabolites, such as sterols, and informative fossilization products of biopolymers, demonstrates the significance of siderite (iron carbonate) concretions in the exceptional preservation of biomolecular information in deep time, facilitated by the rapid encapsulation and remineralisation of organic matter within days to months.

**Abstract:**

The reconstruction of ancient trophic networks is pivotal to our understanding of ecosystem function and change through time. However, inferring dietary relationships in enigmatic ecosystems dominated by organisms without modern analogues, such as the Carboniferous Mazon Creek fauna, has previously been considered challenging: preserved coprolites often do not retain sufficient morphology to identify the dietary composition. Here, we analysed *n* = 3 Mazon Creek coprolites in concretions for dietary signals in preserved biomarkers, stable carbon isotope data, and macromolecular composition. Cholesteroids, metazoan markers of cholesterol, show an increased abundance in the sampled coprolites (86 to 99% of the total steranes) compared to the surrounding sediment, indicating an endogenous nature of preserved organics. Presence of unaltered 5α-cholestan-3β-ol and coprostanol underline the exceptional molecular preservation of the coprolites, and reveal a carnivorous diet for the coprolite producer. Statistical analyses of *in situ* Raman spectra targeting coprolite carbonaceous remains support a metazoan affinity of the digested fossil remains, and suggest a high trophic level for the coprolite producer. These currently oldest, intact dietary stanols, combined with exquisitely preserved macromolecular biosignatures in Carboniferous fossils offer a novel source of trophic information. Molecular and biosignature preservation is facilitated by rapid sedimentary encapsulation of the coprolites within days to months after egestion.

## 1. Introduction

The reconstruction of ancient trophic networks is pivotal to our understanding of ecosystem function and change through time. However, inferring dietary relationships in enigmatic ecosystems dominated by organisms without modern analogues, such as the Carboniferous Mazon Creek fauna, has previously been considered challenging. Coprolites, fossil faecal materials, can offer unique insights into the diets and trophic relationships of extinct life forms in deep time. However, fossil faecal matter can be difficult to interpret, due to the digestion and thus substantial degradation of organismal morphologies. Coprolites can be linked to a producer based on their shape, mineralogy, and geological context of the specimen as well as the presence of any identifiable remains, e.g., [1,2,3,4,5,6,7]. Recently, methods of identification have been expanded to include ^13^C/^12^C, ^15^N/^14^N and DNA analysis [8,9]. However, many coprolite specimens still remain of ambiguous origin and composition and may not contain DNA remains, especially in samples from deep time.

The preservation of biomolecules in deep time is primarily dictated by the early diagenetic chemo-environment, e.g., [10,11,12]. Delicate details in soft tissues are only known from Konservat Lagerstätten [13,14,15,16]; Lagerstätten that preserve soft tissues offer generally more representative insights into extinct biodiversity than fossil sites biased towards the preservation of only hard tissues. The conditions, which result in soft-tissue preservation, are often conducive to the preservation of detailed molecular information, which is otherwise lost during heterotrophic reworking prior to preservation. Carbonate concretions are known to frequently contain carbonaceous fossil soft tissues, e.g., [17], and are thus inferred to form through rapid authigenic mineralisation, which halts significant destruction of the specimen [14,18,19]. During fossilization, lipids are transformed into stable derivatives preserving their original hydrocarbon skeleton. The resulting biomarkers are commonly used to identify original sources of organic matter input into sediments [20].

Biomolecules such as sterols are present in most eukaryotes and contain structural features specific to their function within a group of organisms [21]. Lipids such as sterols tend to be better preserved in sediments compared to other classes of biomolecules (e.g., carbohydrates, proteins) which are often completely degraded under anoxic conditions [22]. Through early diagenesis, biological steroids are defunctionalised as a result of the activity of microbes and clay catalysts, while typically retaining most of their isomeric characteristics [23]. These compounds can include stanols, sterenes, diasterenes and A/B/C-ring monoaromatic steroids. Further diagenetic and catagenetic reactions will result in isomerisation and aromatisation to form more thermodynamically stable configurations [23,24,25], primarily steranes and triaromatic steroids, which can reside in the sediments for up to hundreds of millions of years. A diagenetic continuum of such steroid hydrocarbons was identified in a study of a Devonian calcite concretion from the Gogo Formation, an exceptional Konservat Lagerstätte in the Canning Basin of Western Australia, by Melendez et al. [26]. Here, microbially mediated eogenetic processes were determined to have resulted in the parallel preservation of stenols and their diagenetic products encompassing steranes and triaromatic steroids. Such instances of preservation offer the opportunity to study not only the characteristic biomolecules from the fossilised specimen, but also the post-depositional processes, which transform these biomolecules and the broader taphonomic history of coprolites.

In contrast, structural biomolecules shared among all organisms, such as proteins, tend to crosslink with oxidation products of lipids and sugars via advanced glycoxidation and lipoxidation reaction schemes to form insoluble complex organic matter composed of heteroatom-rich polymers. The resulting insoluble organic matter retains chemically altered, but not unrecognisable evidence of original biosignatures. 

This compositional complexity and potential for chemical transformation through digestive and diagenetic processes requires the combination of complementary chemical analyses. Coprolites have been known to preserve lipid biomarkers, e.g., [27,28,29,30,31], which can be used to reconstruct various molecular inputs including direct dietary information and the processes, which alter these molecules through the digestive tract of the producer [28,32]. In studies of faecal samples, 5β-stanols are common reduced products of dietary sterols including cholesterol, campesterol, sitosterol and stigmasterol [27]. There is a predominance of 5β-cholestan-3β-ol (coprostanol) in human faeces [33], while herbivorous mammal faeces contain mostly 5β-campestanol and 5β-stigmastanol [27,33]. 

In addition, insoluble organic matter resulting from the diagenetic crosslinking of structural biomolecules has been shown to preserve biologically informative heterogeneities for fossils from the Mazon Creek locality [34]. Extracting dietary information from molecular biomarkers and biosignatures preserved in coprolites opens up new opportunities for tracing trophic networks through time. For example, the presence of high amounts of cholestane in coprolite samples has been attributed to a predominantly carnivorous or omnivorous diet, e.g., [28,29], due to ubiquity of cholesterol in animal tissue. In contrast, the presence of an array of phytosterols and long-chain *n*-alkanes derived from leaf waxes can indicate a herbivorous diet, e.g., [29,33,35]. A fraction of the organic material is chemically transformed into insoluble complex organic matter following ingestion, digestion and egestion, potentially preserving dietary tissue information in form of informative heterogeneities. On the other hand, primary dietary information can also be obtained from indigestible components, which pass through the gastrointestinal system without chemical modification.

The accuracy and detail of the molecular dietary reconstruction is reliant on the degree of biomolecule and biomarker preservation, and diagenetic transformation. The processes, which are responsible for preservation of soft tissue (e.g., rapid burial, rapid mineral growth [14]) are also those which have been demonstrated to preserve biomarkers of low maturity and intact biomolecules, e.g., [11,12,26]. Therefore, exceptionally preserved coprolites that were sealed from environmental influence during earliest diagenesis, such as those from the Mazon Creek Lagerstätte, are likely to preserve biomolecular information alongside soft tissues and thus are ideal candidates for ancient dietary reconstructions.

### Palaeoenvironmental Setting

The Mazon Creek Konservat Lagerstätte is one of the most productive fossil Lagerstätten worldwide, with over 350 species of plants and 465 of animals identified [36,37]. The site is renowned for its preservation of delicate soft tissue fossils within iron carbonate (siderite) concretions, of which there are many hundreds of thousands collected. Palynological and palaeobotanical studies of the Lagerstätte, e.g., [36,38,39] have determined its age to be Middle Pennsylvanian (Westphalian D) (306–311 Ma). The siderite concretions are found in the lower 3–8 m of the Francis Creek Shale Member of the Lower Carbondale Formation, overlying the Colchester Coal (No. 2) Member [40,41,42,43,44]. The Mazon Creek site was interpreted as representing a river delta system, e.g., [41,42,43,44,45], wherein some large-scale events such as storms or flash flooding caused the rapid burial of massive amounts of organisms, e.g., [42]. More recently the palaeoenvironment has been re-evaluated to represent a low energy, brackish marine environment where under anaerobic conditions input from peaty forests provided a means of rapid burial of organisms [43]. The abundance of siderite concretions and its occasional co-occurrence with pyrite indicates minimal marine input and low sulfate concentrations, where iron initially reacted with H_2_S (where available) to form pyrite. Under anoxic conditions the presence and activity of methanogenic archaea and methanotrophic bacteria initiated the precipitation of iron carbonate, e.g., [42,46,47,48]. This has also been supported in geochemical studies of ^34^S/^32^S, ^13^C/^12^C and ^18^O/^16^O isotopic compositions of the Mazon Creek concretions [43]. Preservation occurs during early microbial decay of deposited organisms, wherein a ‘proto-concretion’ is formed by the breakdown of degrading organic matter into fatty acids, resulting in mineral precipitation of primarily siderite in an iron-rich environment with limited sulfate [43]. Growth and decay experiments, e.g., [49,50,51,52] have demonstrated that this process is initiated within the weeks after deposition. Studies [53,54] have identified that these processes occurred rapidly (within weeks to months) in carbonate concretions formed around decaying soft tissues of tusk-shells.

Here, we analyze *n* = 3 Mazon Creek coprolites preserved in carbonate concretions for dietary signals in preserved biomarkers, stable carbon isotope data, insoluble fossil organic matter composition, and preserved morphology, to provide comprehensive and complementary insights into the trophic structure of the enigmatic Mazon Creek ecosystem and improve our understanding of the role of concretion formation in the preservation of biomolecules in deep time.

## 2. Materials and Methods

### 2.1. Sample Preparation and Extraction

Three coprolite fossils in siderite concretions were prepared for analysis (Figure 1). The samples used in this study were obtained from the Chicago Field Museum. Samples had previously been collected from the Pit 11 coal strip mine in Kankakee County, Illinois, from the Francis Creek Shale Member of the Carbondale Formation.

One half of each concretion was cut using a handheld Dremel rotary tool with a diamond blade (previously cleaned by sonicating in a mixture of dichloromethane (DCM) and methanol (MeOH) (9:1 *v*/*v*) for 15 min intervals, separating the central coprolite fossil (referred to herein as the ‘fossil’) from the surrounding rock (‘matrix’) for each sample. The different sections of each concretion were washed by repeated sonication in 15 min intervals to trace removal of external contamination in a mixture of DCM:MeOH (9:1 *v*/*v*), before being ground using pre-annealed (450 °C for 3 h) ceramic mortars and pestles. In between sample treatments, pre-annealed sand was ground to clean mortars. 

The ground sample material was Soxhlet extracted in individual pre-extracted cellulose thimbles (Soxhlet extracted three times for 24 h using a mixture of DCM:MeOH (9:1 *v*/*v*)) for 72 h. A procedural blank of a pre-extracted thimble was run alongside each extraction. The samples were filtered through activated copper powder to remove elemental sulfur. Small scale column chromatography (5 cm × 0.5cm i.d.) using silica gel activated at 160 °C for 24 h was used to separate the total lipid extracts into aliphatic (4 mL *n*-hexane), aromatic (4 m *n*-hexane:DCM (9:1 *v*/*v*)), porphyrin (4 mL *n*-hexane:DCM (7:3 *v*/*v*)) and polar (4 mL DCM:MeOH (1:1)) fractions for analysis.

### 2.2. Gas Chromatography-Mass Spectrometry (GC-MS)

Full scan gas chromatography-mass spectrometry analysis (GC-MS) was performed on the aliphatic fractions using an Agilent 7890B GC with a DB-1MS UI capillary column (J and W Scientific, 60 m, 0.25 mm i.d., 0.25 μm film thickness) coupled to an Agilent 5977B MSD. Aromatic fractions were analysed on an Agilent 6890N GC with a DB-5MS UI capillary column (J and W Scientific, 60 m, 0.25 mm i.d., 0.25 μm film thickness) coupled to an Agilent 5975B MSD. The GC oven was ramped from 40 °C to 325 °C at a rate of 3 °C/min with initial and final hold times of 1 min and 30 min, respectively. 

Saturated and aromatic steroids and hopanoids were quantified using GC-MS analyses on an Agilent 6890 GC coupled to a Micromass Autospec Premier double sector MS (Waters Corporation, Milford, MA, USA). The GC was equipped with a 60 m DB-5 capillary column (0.25 mm i.d., 0.25 μm film thickness; Agilent J&W Scientific, Agilent Technologies, Santa Clara, CA, USA), and helium was used as the carrier gas at a constant flow of 1 mL/min. Samples were injected in splitless mode into a Gerstel PTV injector at 60 °C (held for 0.1 min) and heated at 260 °C min^−1^ to 300 °C. The MS source was operated at 260 °C in EI mode at 70 eV ionization energy and 8000 V acceleration voltage. All samples were injected in *n*-hexane to avoid deterioration of chromatographic signals by FeCl_2_ build-up in the MS ion source through use of halogenated solvents [55]. The GC oven was programmed from 60 °C (held for 4 min) to 315 °C at 4 °C min^−1^, with a total run time of 100 min. Saturated steranes and hopanes were quantified using metastable reaction monitoring (MRM) in M^+^ → 217 and M^+^ → 191 transitions, respectively. Mono- and triaromatic steroids were detected using selected ion recording (SIR) under magnet control of base ions *m*/*z* 253 and 231, respectively. All ratios and abundance proportions are reported uncorrected for differences in MS-response. Saturated and aromatic steroid hydrocarbons as measured using MRM are shown in Figure 2 and Figure 3.

### 2.3. Gas Chromatography-Isotope Ratio-Mass Spectrometry (GC-irMS)

Stable isotope ratio mass spectrometric analyses of individual compounds were performed on the aliphatic fractions of each sample to determine δ^13^C, using a Thermo Scientific Trace GC Ultra coupled to a Thermo Scientific Delta V Advantage mass spectrometer via a GC isolink and a Conflo IV. The reactors consisted of a combustion interface containing a ceramic tube lined with NiO and filled with NiO and CuO, held at 1000 °C. The programs used for the GC column, carrier gas, injector conditions and oven temperatures were identical to those used for GC-MS analysis as described above. The gas chromatography-isotope ratio-mass spectrometer (GC-irMS) measures the δ^13^C values by monitoring the CO_2_ produced by the sample and measuring the response of ions of *m*/*z* 44, *m*/*z* 45 and *m*/*z* 46, relative to the reference gas of known δ^13^C content.

### 2.4. Bulk Stable Carbon Isotopes

Ground residue from lipid biomarker extractions were treated with hydrochloric acid (4M) to remove carbonate mineral via repeated addition of fresh acid solution, stirring and heating at 50 °C until gas production ceased. Samples were subsequently washed with Milli-Q water and freeze-dried to remove excess water. δ^13^C analyses were performed using a Thermo Flash 2000 HT elemental analyser (EA) connected to a Delta V Advantage isotope ratio monitoring mass spectrometer (irMS) via a Conflo IV. Samples were weighed (approximately 6 mg) in triplicate into tin cups (SerCon) and combusted to CO_2_ in the nitrogen-carbon reactor (1020 °C). CO_2_ passed through the Conflo IV interface into the irMS, which measured *m*/*z* 44, 45 and 46. δ^13^C values were calculated by Thermo Isodat software and normalised to the international VPDB scale by multi-point normalisation using the standard reference materials NBS 19 (+1.95‰) and L-SVEC (−46.60‰) [56]. The standard reference material IAEA-600 was measured to evaluate the accuracy of the normalization. The normalized δ^13^C values of IAEA-600 from these measurements were within ±0.1‰ of the reported value of −27.77‰ [56].

### 2.5. Polar Compound Analysis

Polar fractions were analysed at Leeder Analytical (Victoria, Australia). Fractions were dried and internal standard (^13^C-cholesterol) added, and were then combined with *N*,*O*-bis(trimethylsilyl)fluoroacetamide and trimethylchlorosilane (99:1) and heated (60 °C) for 20 min. Samples were dissolved in toluene (500 μL) before analysis. Gas chromatography-tandem mass spectrometry (GC-MS/MS) was performed on an Agilent 7890B Gas Chromatograph with a DB-5MS UI column (30 m × 0.25 mm 0.25 um film) coupled to an Agilent 7000D Triple Quadruple Mass Spectrometer. Results were quantified against sterol trimethylsilyl-derivative standards. 

### 2.6. In Situ Raman Microspectroscopy and ChemoSpace Analysis of Spectral Data 

The set of analysed coprolites was microscopically screened for evidence of preserved carbonaceous matter characterized by a dark brown-to-black colouration (Figure 1), and two coprolites (FMNH PE 52316, FMNH PE 52336) with suitable preservation were identified. A total of *n* = 35 carbonaceous vertebrate, annelid, non-annelid invertebrate, and plant fossils from the Mazon Creek locality (Appendix A) and the two carbonaceous coprolites (FMNH PE 52316, FMNH PE 52336) were surface-cleaned with 70% EtOH, and subjected to high-resolution *in situ* Raman microspectroscopy in the Department of Earth and Planetary Sciences at Yale University. Raman microspectroscopy was performed using a Horiba LabRam HR800 with 532 nm excitation (holographic notch filter; 20 mW at the sample surface). The spectra were obtained in LabSpec 5 software, and the instant processing included only a standard spike removal. Raman scattering was detected by an electron multiplying charge-coupled device (EM-CCD) following dispersal with an 1800 grooves/mm grating and passing through a 200 μm slit (hole size 300 μm). The spectrometer was calibrated using the first order Si band at 520.7 cm^−1^. Ten spectra were accumulated in the 500–1800 cm^−1^ region, also known as ‘organic fingerprint region’, for 10 s exposure time each, at 32× magnification. All spectra were analyzed in an identical fashion in SpectraGryph 1.24 spectroscopic software: A conservative adaptive baseline (30%) was fitted, no baseline offset was imposed, and all spectra were normalized to the common highest peak. Relative intensities (*n* = 53; arbitrary units) at pre-selected informative band positions ([34,57]; listed in the Appendix A) were exported using the ‘Multicursor’ function in SpectraGryph 1.2. The resulting variance-covariance matrix was exported into PAST 3 (file available as Source Data), and two separate sample identification strategies were applied: One data matrix contains the extracted spectroscopic signals and binary characters identifying carbonaceous plant, vertebrate, annelid, non-annelid invertebrate remains and coprolites as separate tissue types. A second matrix uses a more agnostic approach and contains, in addition to the spectroscopic signals, only binary characters identifying samples as plants, annelids, non-annelid invertebrates, and vertebrates; in this data matrix coprolite tissue affinity was coded as ‘unknown’. Both data matrices are available as Source Data. Endogeneity of organic matter in the carbonaceous films associated with fossil morphology has previously been demonstrated [57], and is here separately assessed using lipid biomarkers. A Canonical Correspondence Analysis of the first data matrix revealed the diagnostic molecular features distinguishing the fossil coprolites from other fossil soft tissues from the Mazon Creek locality (Figure 4A). A second Canonical Correspondence Analysis allowed the coprolite samples to locate in the compositional space (ChemoSpace) based on the affinity of the digested, fossil tissues (Figure 4B). Canonical Correspondence Analysis is a discriminant multivariate analysis that distinguishes previously identified groups of samples in a ChemoSpace, and, if samples of unknown affinity are included, reveals their affinity (which is here translated into the dietary spectrum of a coprolite producer). Due to the discriminant, comparative nature of the analysis and spectroscopic data, axis loadings do not have a dedicated unit. The impact of all *n* = 53 extracted relative intensities is represented by colour-coded (see caption) ChemoSpace vector arrows in Figure 5 (Figure 5B corresponding to the ChemoSpace shown in Figure 4A, and Figure 5C corresponding to the ChemoSpace shown in Figure 4B). Functional groups were identified using Lambert et al. [58], and have been previously published [34,57]; select functional groups that are characteristically enriched in coprolites and reveal the affinity of their digested tissues are labelled in Figure 5. 

### 2.7. X-ray Diffraction 

Powdered samples were analysed using a Bruker-AXS (Karlsruhe, Germany) D8 Advance Powder Diffractometer with a CuKα radiation source (40 kV, 40 mA) and a LynxEye detector. The scan ranged from 5° to 90° 2θ with a step size of 0.015° and a collection time of 0.7 s per step. Crystalline phases were identified by using the Search/Match algorithm, DIFFRAC.EVA 5.2 (Bruker-AXS, Karlsruhe, Germany) to search the International Center for Diffraction Data (ICDD) Powder Diffraction File (PDF4+ 2021 edition).

### 2.8. Elemental Analysis

Elemental analyses were performed at Source Certain International (Western Australia). Samples (0.25 g) were accurately weighed and digested in nitric acid (16 mL, 65 wt%), perchloric acid (4 mL, 70 wt%) and hydrofluoric acid (10 mL, 50 wt%) at approximately 180 °C for a minimum of 16 h under reflux. The acids were removed by evaporation at approximately 220 °C. Once the residue reached incipient dryness it was dissolved in hydrochloric acid (0.75 mL, 32 wt%), nitric acid (0.25 mL, 65 wt%) and DI-water (20 mL, >16.4 MΩ cm). The solution was suitably diluted for the instrument. Samples were analysed using an Agilent 5110 ICP-AES and an Agilent 7700 ICP-MS. 

### 2.9. Total Organic Carbon

The rock samples were ground to a fine powder and digested in acid (HCl) to remove the carbonate minerals. The remaining residues were analysed using a LECO Carbon-Sulfur Analyser (CS-230). The CO_2_ produced was measured with an infra-red detector, and values calculated according to standard calibration.

## 3. Results and Discussion

Results are discussed in general terms of all three samples, unless a particular sample is specified. Samples subject to the same analysis showed generally consistent results as reflected in biomarker ratios presented in Table 1, Table 2, Table 3 and Table 4. Figures present the sample which best demonstrates compositional features. 

### 3.1. Inorganic Composition

The coprolite components of samples PE 52316 and PE 52336 are approximately 2 cm and 3.5 cm in length, respectively and appear to be similar in composition (Figure 1). Three primary mineral regions were identified by X-ray diffraction (XRD). The coprolites are preserved in three dimensions as calcium phosphate with cracks filled with sphalerite. The concretionary material is siderite with minor amounts of quartz. Total organic carbon ranged from 0.31 to 0.78 wt% of rock in the fossils and 0.32 to 0.51 wt% in the matrices.

Phosphatic preservation is a characteristic component of carnivore coprolites, e.g., [1,4,59]. It has been suggested that rapid precipitation (within weeks—e.g., [14,19,60]) of dietary calcium phosphate can result in the preservation of fine morphological information [4,61]. Much like the formation of carbonate concretions, the remineralisation of calcium phosphate from faecal material has been demonstrated as autochthonous [7], occurring rapidly after deposition and prior to diagenetic permineralisation. 

Elemental analyses revealed enrichment of rare earth elements (REE) in the coprolite fossils compared to the siderite concretion. Phosphatic minerals such as apatite are able to incorporate REE during early fossilisation [62] via substitution for calcium, e.g., [63]. REE are commonly used in palaeontological, palaeoenvironmental and palaeoredox studies, particularly those focused on vertebrate bones, e.g., [64,65,66,67,68] as well as on coprolites fossilised as apatite, e.g., [69,70]. 

### 3.2. Lipid Biomarkers of Coprolites

The coprolite fossil regions are characterised by a predominance of cholesterol-derived steroidal hydrocarbons (e.g., Figure 2 and Figure 3, Table 1). C_27_ cholestanes make up 86 to 99% of the total steranes composition of the aliphatic fraction (Table 1). Similarly, aromatic cholesteroids make up the majority of the monoaromatic and triaromatic steroid distributions of each of the fossils (Figure 3, Table 1). Cholesterol and its derivatives have been considered in past studies of faecal material as indicators of an animal diet, e.g., [28,29]. Cholesterol is generally known to be synthesised by animals, while ergosteroids can be found in fungi and some groups of algae [71,72] and sitosterol and stigmasterol are commonly made by higher plants [71]. As such, the relative proportions of related sterane biomarkers can help distinguish the contributions of these inputs in sedimentary organic matter input [73]. However, cholesteroids can also be produced by herbivores via modification of other sterol analogues, particularly phytosterols [74] or synthesised *de novo*, e.g., [74,75,76] and are also present in almost all eukaryotic cell walls [77]. While the exceptionally high abundance of cholesteroid biomarkers suggests a primarily animal diet for the coprolite producer, minor contributions from plant material as a dietary component or due to occasional grazing cannot be completely ruled out. 

The diagenetic fate of biomarkers is typically controlled by factors such as burial depth, redox conditions, mineral and porewater chemistry and geological time [20]. The low diagenetic transformation of the cholesteroids in the coprolite specimens is likely related to favourable preservation conditions during aromatisation. Similarly immature biomolecular signals have been found in previous studies of carbonate concretions where rapid encapsulation, burial and mineralisation supported the preservation of soft tissues and intact biomolecules, e.g., [10,12,26].

Diagenetic processes promote rearrangement of ααα 20*R* stereoisomers to the more thermodynamically stable ααα 20*S* and αββ 20*R* + *S* compounds [20,25]. Within the cholestanes distribution the biologically derived cholestane 20*R* isomer is most dominant (e.g., Figure 2A). Additionally, present is C_27_ ααα 20*S* cholestane coeluting with coprostane (C_27_ βαα 20*R*). Here, the predominance of the C_27_ ααα 20*R* over the 20*S* isomer, (i.e., C_27_ ααα 20*S*/(20*S* + 20*R*) < 0.2), support a low diagenetic conversion of the steroids inside the fossil. 

In contrast, the side chain isomerisation of C_27_ to C_29_ steranes as well as the transformation of ααα to αββ-steranes progressed much further in the concretion matrix (Figure 2). This indicates that diagenetic conversion of labile steroids was hampered by early cementation and the lack of clay mineral catalysis, e.g., [11,78,79] in the coprolite, allowing the labile steroids to persevere.

Due to coelution, the proportion of βαα 20*R* compared to ααα 20*S* was not precisely determined; however, it is presumed that a large area of this peak comprises C_27_ βαα 20*R* based on the low diagenetic conversion of steroids within the fossil and generally low maturity. This was confirmed by the mass spectrum which showed the presence of two coeluting peaks, one with *m*/*z* 149 fragment (C_27_ αββ 20*S*) and the other with *m*/*z* 151 fragment (C_27_ βαα 20*R*). Steroids with βαα stereochemistry can be produced by reduction of sterols in the intestine of mammals depending on the primary steroid components of their diet [27] or may also form in sediment through microbial reduction of ∆^5^ sterols [80]. 

∆^2^-sterenes and ∆^3,5^-steradienes are products formed in early diagenesis via dehydration of 5α(H)-stanols and ∆^5^-sterols, respectively [23,81]. ∆^2^-sterenes can also undergo isomerisation and rearrangement to ∆^4^- and ∆^5^-sterenes and subsequently to diasterenes [23,24,82,83]. A cholestadiene compound has been tentatively identified by comparison of its mass spectrum with that of a 3,5-cholestadiene [81], based on predominant fragments at *m*/*z* 213 and 368, in sample PE 52336. This compound is present in a higher relative abundance in the fossil than in the matrix, suggesting that it is likely derived from the original cholesterol content in the coprolite sample. Diasterenes in low abundance were also identified, although in the fossil of PE 52336 only, and are shown in the Appendix A.

A- and B-ring monoaromatic steroids are derived directly from sterol compounds during early diagenesis, e.g., [84,85,86] while C-ring monoaromatic steroids have previously been attributed to later-stage diagenesis, e.g., [87], which can then be aromatised to form triaromatic hydrocarbons via loss of a methyl group, with increasing thermal maturity [88]. However, it has been suggested, e.g., [26,89] that aromatisation of sterene hydrocarbons to C-ring monoaromatic and triaromatic steroids can occur during early diagenesis via microbially mediated processes. Both the monoaromatic and triaromatic steroids show abundant cholesteroids compared to ergosteroids and stigmasteroids, with cholesteroids comprising between 55 to 82% of the total monoaromatic steroids distribution and 43 to 94% of total triaromatic steranes (Table 1). A predominance of the cholesteroid analogues in all steroid classes demonstrates that diagenetic transformation, while altering the cholesterol predominance somewhat, does not eliminate the primary dietary composition of steroids. Based on conventional models of triaromatic steroid formation, presence of triaromatic steroids in the fossil suggests a catagenetic history, which is inconsistent with the immaturity of the sample. It is therefore plausible that triaromatic steroids were also formed by microbially mediated reaction mechanisms, e.g., [26,89].

5α-cholestan-3β-ol was identified in the fossil of PE 52336 and PE 52316 (Table 2). This stanol may be either from a direct biological input or from early diagenesis by reduction of cholesterol [23,90]. The presence of 5α-cholestan-3β-ol in several of the fossils and its absence in any of the matrices supports that this is derived from the fossil coprolite itself. Coprostan-3-ol (5β-cholestan-3β-ol) was identified, *albeit* in low abundance in the PE 52315 fossil only (Table 2). This is a possible precursor sterol of coprostane that can be formed via reduction of cholesterol in the gut of many higher mammals, e.g., [27] as well as through microbial reduction in sediment [80]. Cholesterol was also identified in the fossil and matrix of all samples; however, this must be considered with caution, as it is also present in comparable concentrations in the procedural blank (Table 2).

### 3.3. Early Diagenetic Transformation of Dietary Sterols

A series of diagenetic products derived from cholesterol, such as cholestane and triaromatic steroids are preserved in all fossil coprolites. These components support a primarily carnivorous dietary source as they would support a high amount of cholesterol in the original faecal material, which has been rearranged and partially preserved. The cholesteroid compounds identified are summarised in Figure 6. Cholesterol can undergo reduction to 5α- and 5β-stanols [23], such as 5α-cholestan-3β-ol as present in the fossil coprolites. Further reduction would yield cholestane; specifically, the 5α-cholestane, which is the abundant sterane identified in each of the fossils. Cholesterol can also undergo reduction directly to Δ^2^-sterenes and Δ^3,5^-steradienes [23]. While sterenes were not observed, a single cholestadiene was identified in the PE 52336 fossil, which is formed from cholesterol and may be further converted into monoaromatic (Figure 3A) and triaromatic (Figure 3B) steroids.

### 3.4. Raman ChemoSpace

The Raman ChemoSpace is here used to complement compositional insights from the analysis of biomarkers with additional information on the molecular makeup and preserved biosignatures in the macromolecular fraction of Mazon Creek coprolites (FMNH PE 52316, FMNH PE 52336). Focus of the analysis are *in situ* Raman spectra collected for carbonaceous fossil remains. The discriminant analysis (CCA) of spectral data collected for all *n* = 37 samples, shown in Figure 4A, identifies key compositional characteristics of fossil vertebrate, non-annelid invertebrate, annelid, plant, and coprolite samples from the Mazon Creek locality. Among these samples, only minimal overlap between the clusters of the diverse array of fossil metazoan tissues is observed; the cluster of carbonaceous plant tissues is separated from the metazoan tissue clusters (Figure 4A). Using the spread of samples in the compositional space (=ChemoSpace), the circular fractions including all metazoan (orange) and plant (green) tissues were plotted in the CCA (Figure 4A,B). It is the nature of the CCA that separates different tissue categories (black vector arrows) radiating from the origin into the ChemoSpace, therefore allowing to constrain the distribution of samples through circular fractions. The trajectories of all tissue vectors and their associated, characteristic molecular features are plotted in Figure 5. Figure 4A locates the two coprolite samples within the circular fraction occupied by metazoan tissues, and reveals as characteristic molecular features a relatively increased abundance of alcohol (C-OH) and carbonyl groups (C=O), aromatics, inorganic phosphate (P-O, PO_4_^3-^), and organo-sulfur moieties (C-S; Figure 5B). The majority of these characteristic molecular features that discriminate fossil coprolites from undigested fossil tissues coincide with the key chemical modifications of macromolecules experienced prior to fossilisation, during *in vivo* enzymatic digestion (Figure 5A): enzyme- and acid-catalysed (HCl_(aq)_, stomach acid) hydrolysis cleaves primarily peptide bonds (amides), glucosidic bonds, esters (including phosphoesters), and breaks down lipids into their aliphatic and aromatic building blocks (please see steroid and hopanoid analyses in this study). Hydrolytic cleavage introduces alcohol groups, which can be converted into carbonyls during subsequent alteration. The molecular fingerprint of hydrolytic digestion preserves in the sampled Mazon Creek coprolites even after oxidative alteration (crosslinking reactions) acting during fossilisation [34,57]. Digestion moves data points in the CCA ChemoSpace along the vectors associated with functional groups introduced during hydrolytic cleavage (colored vectors in Figure 5B,C). The relative increase in organo-sulfur moieties is not related to digestive hydrolysis and thus relates either to the gastro-intestinal concentration processes or post-egestion alteration. 

Figure 4B shows a complementary CCA ChemoSpace which, contrary to the analysis in Figure 4A, did not treat coprolites as a separate tissue category, but instead allowed them to group with the other *n* = 4 tissue categories based on the affinity of the contained, digested remains. Both coprolite samples plot within the circular fraction occupied by metazoan samples, and one tangentially overlaps with the vertebrate tissue cluster. Considering that digestion shifts samples in the ChemoSpace CCE (shown in Figure 4A and Figure 5B,C) towards the right, we thus infer a metazoan affinity of the digested faecal matter and identify the coprolite producer as a carnivore rather than omnivore or herbivore (no primary contribution of plants to the diet). 

The carnivorous diet, potentially based on vertebrates (as suggested by the proximity to the vertebrate cluster in Figure 4A,B), suggests that the producer of the coprolites represented a high trophic level in the ancient Mazon Creek ecosystem and was potentially an apex predator. Future experimental work on the ChemoSpace effects of different metazoan digestive processes has the potential to allow for the detailed quantification of the contributions of vertebrate, plant, non-annelid invertebrate, and annelid food items to the diet of coprolite producers.

The independent data collected for the soluble organic phase (biomarker analysis, stable isotope fractionation) and the insoluble organic phase (ChemoSpace of carbonaceous fossils) converge in their result: a carnivorous diet is inferred for the producers of the analysed coprolites. 

### 3.5. Lipid Biomarkers of Matrix (Palaeoenvironmental Signal)

Terpane and hopane compounds were more dominant relative to steranes in the concretion matrix compared to the coprolite, as reflected in the regular sterane/hopane ratios (1.9–80 in the fossils compared to 0.54–1.1 in matrices) (Table 3). The Ts/(Ts + Tm) ratio, traditionally considered to be indicative of thermal maturity and clay catalysis, indicates the end of the diagenetic or the onset of the catagenetic stage of thermal transformation for both the coprolite and concretion matrix, which stands in contrast to the presence of a thermally labile steradiene and stanol (Table 3). C_29_ 20*S*/(20*S* + 20*R*) steranes ratios (Table 3) are also low overall but consistently higher in the fossil than in the matrix. The Ts/(Ts + Tm) ratios are consistent between the fossil and matrix while the diasterane/regular steranes ratios vary, supporting that different mechanisms are responsible for formation of tris*n**orneo*hopane than for the isomerization of steranes. Dia/regular sterane ratios for all steroid homologs show a higher proportions of rearranged steroids in the matrix than in the fossil. In general, even in the matrix the C_27_ dia/regular sterane ratio indicates lower maturity; however, the C_27_ sterane ratios are likely strongly influenced by source input. 

### 3.6. Stable Carbon Isotopes

Carbon isotopes of the bulk organic matter were measured on the residue of extracted sediment from samples PE 52316 and PE 52336 after treatment with hydrochloric acid to remove carbonate. Both fossil and matrix of each sample showed δ^13^C values around −23.8‰ (Table 4). 

Compound specific carbon isotopes were measured for the *n*-alkanes of these two samples, as well as for the C_27_ ααα 20*S* sterane in the fossil portions (Table 3). δ^13^C values of cholestane (−32.8‰) were depleted by approximately 2.0‰ compared to the average isotopic values of *n*-C_20–25_ (−30.8‰) and *n*-C_18_ (−31.1‰), and by approximately 9.0‰ compared to the bulk organic matter δ^13^C values. 

Sterane hydrocarbons are typically enriched compared to linear hydrocarbons from the same source by up to 8‰ [93]. Cholestane is here interpreted as derived from dietary cholesterol from the coprolite, while slightly more ^13^C enriched *n*-alkanes represent input from external sources, most likely palaeoenvironmental signals. The depletion of δ^13^C value of cholestane in the two coprolites by approximately 2‰ compared to the average of the abundant mid-chain *n*-alkanes is therefore consistent with input from two different sources. The isotopic consistency (approximately −30.8‰) of the *n*-alkanes in the fossil and the matrix is indicative of being derived from a common source, while the depletion of cholestane suggests it has a source different to the *n*-alkanes. The *n*-alkanes in the samples are therefore likely to originate from freshwater producers (e.g., algae, aquatic macrophytes, mosses), while cholesteroids intrinsic to the coprolite derive from animal sources. 

Phytol is also typically ^13^C enriched with respect to fatty acids synthesised in phytoplanktonic cells by 2–5‰ [93]. A similar 2–5‰ distinction between phytane and straight-chain hydrocarbons (and pristane) would therefore be expected of phytane derived from phytol, synthesised by chlorophyll *a* [93]. In these samples phytane was depleted in ^13^C by approximately 2.5–3.5‰ compared to *n*-C_18_ and 2.5–4‰ compared to mid-chain *n*-alkanes (Table 3) which is more typical of an origin within a methane cycle, wherein phytane is derived from ether lipids of methanotrophs and not phytol, e.g., [94,95]. Methanogenic archaea and methanotrophs are considered important components of the microbial growth mechanisms of siderite concretions, e.g., [95]. 

## 4. Conclusions

This study demonstrates that:(1)Cholesteroids including intact 5α-cholestan-3β-ol and coprostanol have been preserved in siderite concretions hosting 306 million-year-old coprolites.(2)The molecular data obtained by GC-MS, GC-MRM, GC-irMS and Raman microspectroscopy supports a primarily carnivorous diet and suggest an elevated trophic position for the coprolite producer.(3)The preservation of intact dietary sterols and macromolecular biosignatures is attributed to rapid encapsulation of the coprolites within days to months after egestion.(4)Siderite (FeCO_3_) concretions (of Carboniferous age) seem to preserve intact and modified, but not unrecognisable, biomolecules much like calcium carbonate concretions of Jurassic and Devonian ages.

The results demonstrate that molecular information preserved within fossils can provide important ancient dietary insights either alongside or independent of traditional mineralogical or morphological studies of coprolites. Intact dietary sterols present in fossils support the significance of rapid encapsulation and organo-templated growth of carbonate concretions in the preservation of biomolecules in geological time. Carbonate concretions, which are host to soft-tissue fossil preservation are evidently important samples for molecular studies and represent each a unique opportunity to study extinct species and past environments.

## Figures and Tables

**Figure 1 biology-11-01289-f001:**
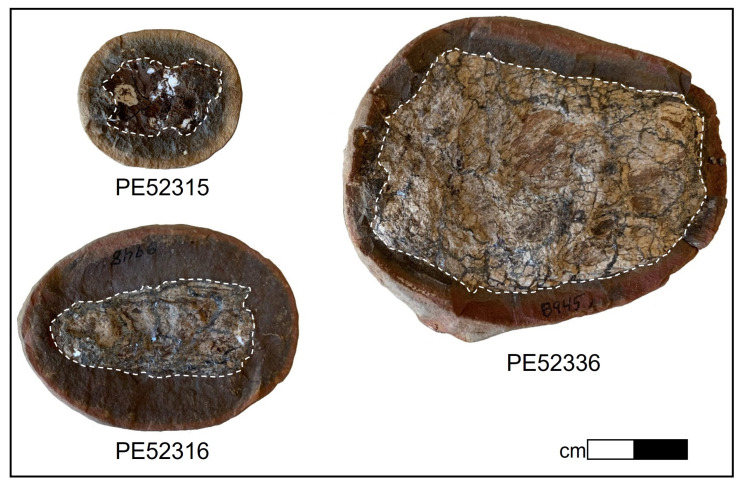
Images of the three samples used in this study. Regions are defined as the ‘Fossil’ and the ‘Matrix’ corresponding to the coprolite fossil region and the concretionary region, respectively, as demonstrated by white dashed lines. Subsamples of each region were used for geochemical analysis. ‘Matrix’ is used throughout to refer to the concretionary region, which is assumed to consist of primarily concretionary material plus potential external organic matter inputs.

**Figure 2 biology-11-01289-f002:**
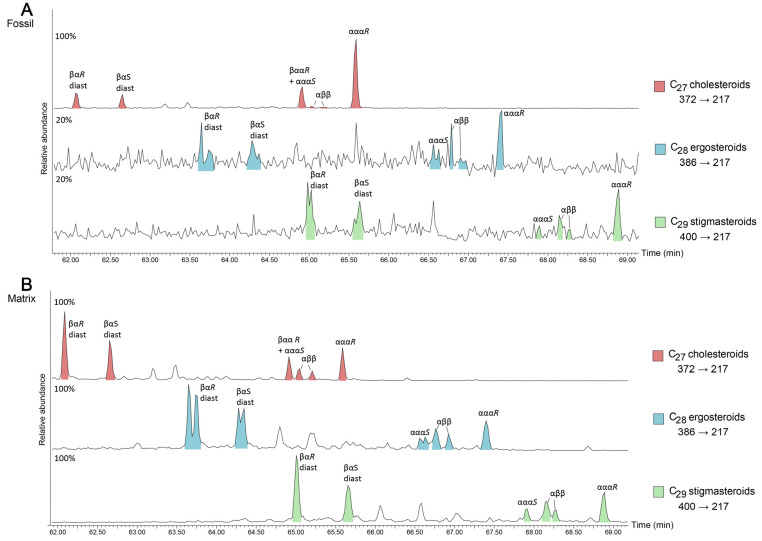
Metastable Reaction Monitoring (MRM) chromatograms of M^+^ → 217 precursor to product transitions of C_27–29_ steranes of PE 52316 Fossil (**A**) and Matrix (**B**). Peaks are coloured according to the number of carbon atoms in the sterane. α and β nomenclature refers to the stereochemistry of hydrogen at C-5, C-14 and C-17 for regular steranes and C-13 and C-17 for diasteranes, while *S*/*R* refers to stereochemistry at the C-20 position. Percentages represent the relative abundance of the most abundant peak in each transition.

**Figure 3 biology-11-01289-f003:**
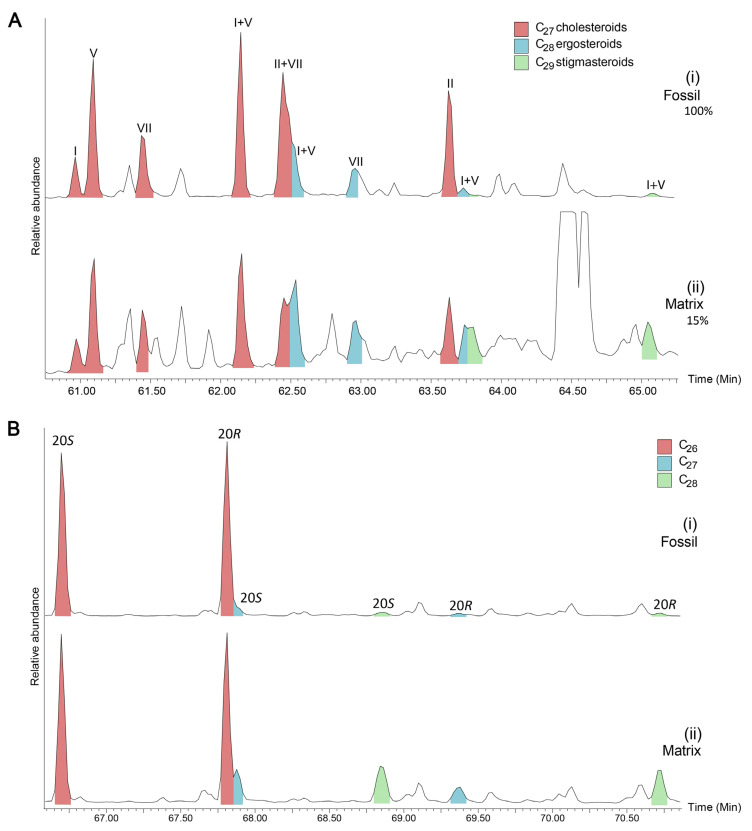
(**A**) Selected Ion Recording (SIR) *m*/*z* = 253 chromatogram of C_27–29_ monoaromatic steroids of PE 52336 Fossil (**i**) and Matrix (**ii**). I = 5β(H),10β(CH_3_); II = 5α(H),10β(CH_3_); V = 5β(CH_3_),10β(H); VII = 5α(CH_3_),10α(H). Percentages represent the relative abundance of the most abundant peak in each transition. (**B**) SIM *m*/*z* = 231 chromatogram of C_26–28_ triaromatic steroids of PE 52336 Fossil (**i**) and Matrix (**ii**). *S*/*R* refers to the stereochemistry at C-20.

**Figure 4 biology-11-01289-f004:**
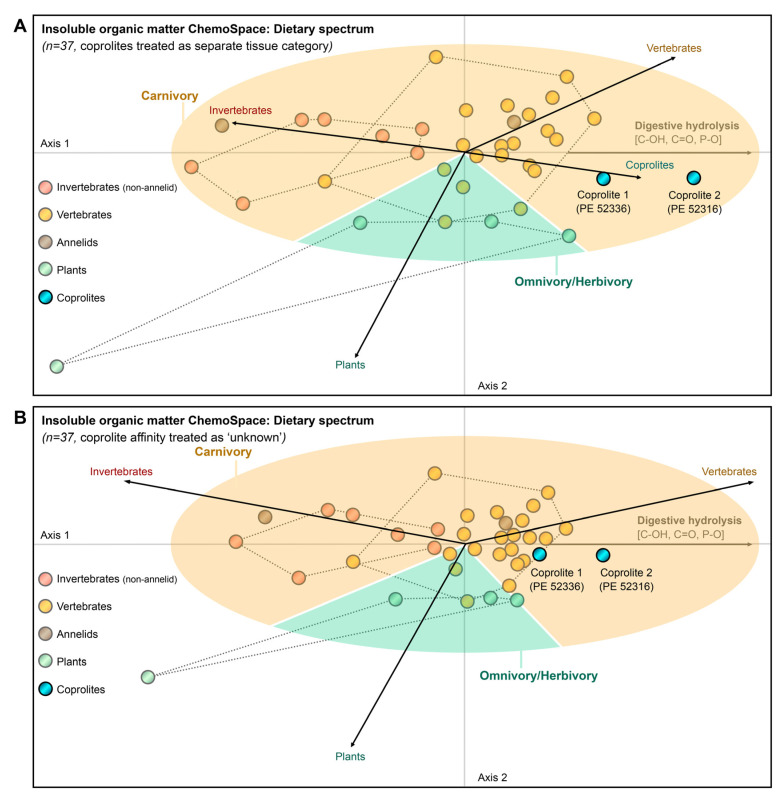
Dietary ChemoSpace analysis of tissue type signals preserved in carbonaceous coprolites from the Mazon Creek locality. A total of *n* = 37 Mazon Creek carbonaceous fossils and coprolites were spectroscopically fingerprinted (*n* = 1 biological replicate and *n* = 10 mean-averaged technical replicates per data point), identified, and analysed by means of a Canonical Correspondence Analysis (CCA; discriminant analysis). (**A**) ChemoSpace resulting from a CCA that treated coprolites as a separate tissue category (black arrows); corresponding trajectories of functional groups in the ChemoSpace are plotted in Figure 5B. (**B**) ChemoSpace resulting from a CCA that treated the tissue affinity of coprolites as unknown; corresponding trajectories of functional groups in the ChemoSpace are plotted in Figure 5C. ChemoSpaces in both, (**A**) (clustered in *n* = 5 categories, each contained in a dotted, convex hull; except from annelids and coprolites due to small sample size) and (**B**) (clustered in *n* = 4 categories, each contained in a dotted, convex hull, except from annelids), reveal that fossil coprolites are compositionally distinct from undigested fossil soft tissues due to digestive hydrolysis of macromolecules prior to fossilisation (detailed differences are plotted in Figure 5B), and contain predominantly digested tissues of vertebrate prey items. Thus, carnivory (orange circle fraction, contrary to omnivory/herbivory = green circle fraction) can be inferred for both coprolite producers. Source Data are available for the CCAs in (**A**,**B**).

**Figure 5 biology-11-01289-f005:**
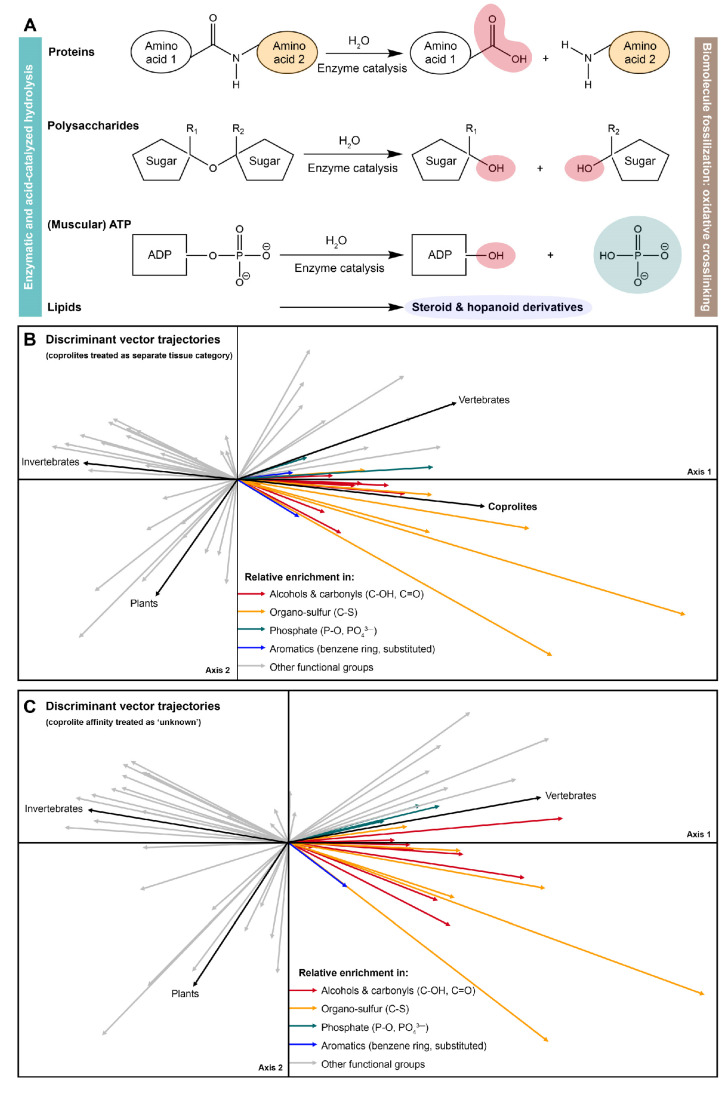
*In vivo* digestive hydrolysis reactions of key macromolecules and functional group trajectories in the compositional spaces plotted in Figure 4. (**A**) Functional group chemistry during digestive hydrolysis reactions of proteins, polysaccharides, adenosine triphosphate (ATP), and lipids: stomach acid and digestive enzymes catalyse the hydrolytic cleavage of amide, glycosidic, and ester bonds and yield a relative increase in the relative abundance of alcohols and carbonyls (red), phosphate (teal), and aromatic compounds (dark blue); Depending on the dietary source, faecal matter can be enriched in organo-sulfur moieties (orange; S-bearing amino acids: cystein, methionine). Compositional differences associated with digestive hydrolysis survive (as shown in Figure 4) biomolecule fossilization through oxidative crosslinking. ADP = adenosine diphosphate; ATP = adenosine triphosphate. (**B**) Discriminant vector arrows for tissue categories (black, *n* = 5) and functional groups (*n* = 53 extracted relative intensities) in the Canonical Correspondence Analysis (CCA) shown in Figure 4A. Functional groups are colour-coded (corresponding to functional group labels in (**A**)) for vectors that explain the ChemoSpace placement of the two analyzed coprolites. This CCA treated coprolites (*n* = 2) as a separate tissue category to reveal the unique compositional features distinguishing them from other types of carbonaceous soft tissues from the Mazon Creek locality (*n* = 35). (**C**) Discriminant vector arrows for tissue categories (black, *n* = 4) and functional groups (*n* = 53 extracted relative intensities) in the CCA shown in Figure 4B. Functional groups are colour-coded (corresponding to functional group labels in (**A**)) for vectors that explain the ChemoSpace placement of the two analysed coprolites (see Figure 4B). This CCA treated coprolites (*n* = 2) as samples with unknown tissue affinity to identify the primary source of fossil faecal matter (*n* = 35).

**Figure 6 biology-11-01289-f006:**
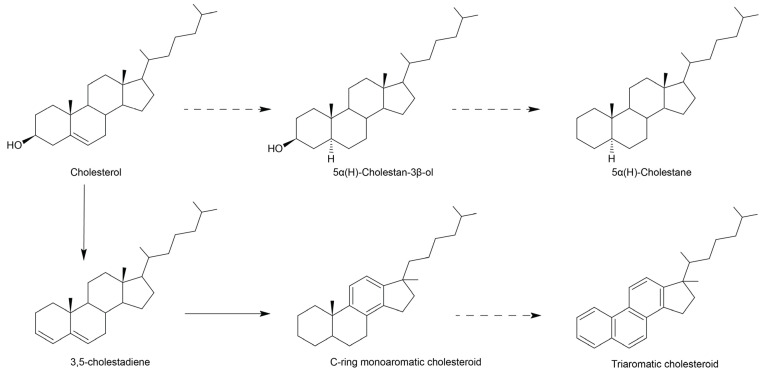
General schematic demonstrating diagenetic rearrangements of cholesteroid compounds, as identified in fossil coprolites in the current study. Dashed lines represent multiple possible intermediates or pathways.In both fossil and matrix of all samples, *n*-alkanes are abundant and consistently range from *n*-C_15_ to *n*-C_26_ with a maximum at *n*-C_23_ but with no odd or even carbon number preference (Figure 7). The predominance of mid-chain (C_20_-C_25_) *n*-alkanes and lack of high-molecular-weight *n*-alkanes supports input from aquatic organic material in a limnic or deltaic environment receiving a minimal input of land plant material, e.g., [91,92]. The freshwater-influenced *n*-alkane distribution is thus in agreement with a geologically inferred large delta setting, e.g., [41,42,43,44,45]. The regular isoprenoids pristane (Pr) and phytane (Ph) are also present with Pr/Ph ratios ranging from 0.85–1.65 (Table 2), ratios supporting fluctuating redox conditions within the environment [73].

**Figure 7 biology-11-01289-f007:**
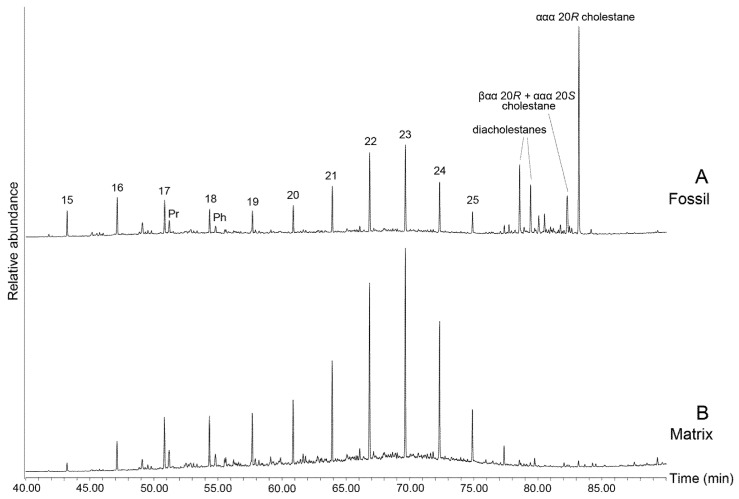
GC-MS total ion chromatogram of PE 52336 showing differences in *n*-alkanes distribution and cholestanes abundances in the Fossil (A) versus the Matrix (B). Numbers represent the carbon number of the hydrocarbon chain. Pr = pristane, Ph = phytane.

**Table 1 biology-11-01289-t001:** Steroid distributions in extracted organic matter.

		PE52315		PE52316		PE52336	
		Fossil	Matrix	Fossil	Matrix	Fossil	Matrix
^1^ Steranes (%)	C_27_	86.2	69.1	92.8	50.1	99.0	78.0
	C_28_	7.3	16.3	3.5	25.0	0.6	9.8
	C_29_	6.5	14.6	3.7	24.9	0.4	12.2
^2^ Reg steranes/hopanes		1.9	0.9	4.8	0.5	80.1	1.1
^3^ Monoaromatic steroids (%)	C_27_	54.8	41.8	68.3	38.9	82.2	56.1
	C_28_	31.7	27.7	25.1	25.9	15.2	22.7
	C_29_	13.5	30.5	6.6	35.1	2.5	21.2
^4^ Triaromatic steroids (%)	C_27_	43.6	26.7	84.6	41.4	94.8	68.4
	C_28_	19.4	27.2	6.1	19.9	2.7	11.5
	C_29_	37.0	46.1	9.3	38.7	2.5	20.1

^1^ Distributions of regular steranes were computed using the sum of regular steranes and diasteranes in MRM M^+^ → 217 transitions. Steranes: C_27–29_ ααα- and αββ-20(*S* + *R*)-steranes and C_27_ βαα 20*R*; ααα = 5α(H),14α(H),17α(H); αββ = 5α(H),14β(H),17β(H); βαα = 5β(H),14 α(H),17 α(H); diasteranes: βα-22(*S* + *R*)-diasteranes; 13β(H),17α(H). ^2^ Reg steranes/hopanes = [∑(C_27–29_ steranes)]/[∑(C_27–35_ hopanes)], steranes: as above; hopanes: C_27_ Ts, Tm 17α, Tm 17β; C_29_ αβ, Ts, βα; C_30_ αβ, βα; C_31–35_ αβ-22(*S* + *R*); αβ = 17α(H),21β(H); βα = 17β(H),21α(H); C_27_ Ts = 18α-22,29,30-tris*norneo*hopane; C_27_ Tm = 17α-22,29,30-trisnorhopane; C_29_ Ts = 18α-30-*Norneo*hopane. ^3^ Distributions of monoaromatic steroids (MAS) were computed using the sum of *S* and *R* I + V isomers of MAS C_27–29_ homologs in *m*/*z* 253 mass chromatograms. ^4^ Triaromatic steroid distributions (TAS) were computed using the sum of C_26–28_ homologs in the *m*/*z* 231 mass chromatograms.

**Table 2 biology-11-01289-t002:** Values of sterol compounds reported as total amount in micrograms (μg) total in each fraction, quantified against external sterol standards in 500 mL, as an average of two measurements.

	PE 52315		PE 52316		PE 52336		Procedural Blank
	Fossil	Matrix	Fossil	Matrix	Fossil	Matrix	
5α-Cholestan-3β-ol	-	-	0.43	-	0.16	-	-
Cholesterol	0.07	0.06	0.05	0.17	0.07	0.14	0.07
Coprostan-3-ol	0.03	-	-	-	-	-	-

**Table 3 biology-11-01289-t003:** Biomarker parameters in extracts of coprolite samples.

	PE 52315		PE 52316		PE 52336	
	Fossil	Matrix	Fossil	Matrix	Fossil	Matrix
^1^ Dia/reg C_27_ steranes	0.64	1.03	0.30	1.50	0.32	0.76
^2^ Dia/reg C_28_ steranes	1.61	2.35	0.74	1.87	0.44	2.09
^3^ Dia/reg C_29_ steranes	1.25	1.52	0.89	1.47	*	1.46
C_29_ 20*S*/(20*S* + 20*R*) steranes	0.28	0.30	0.18	0.27	0.21	0.25
^4^ Ts/(Ts + Tm)	0.52	0.55	0.47	0.52	0.50	0.44
βα/(βα + αβ) C_30_ hopane	0.11	0.10	0.08	0.09	0.09	0.11
^5^ Pr/Ph	1.00	0.99	0.85	1.03	1.65	1.42

^1^ Dia/Reg C_27_ steranes = [∑(βα-22(*S* + *R*)-diacholestane)]/[∑(ααα- and αββ-20(*S* + *R*)-cholestane)]; ^2^ Dia/Reg C_28_ steranes = [∑(βα-22(*S* + *R*)-diaergostane)]/[∑(ααα- and αββ-20(*S* + *R*)-ergostane)]; ^3^ Dia/Reg C_29_ steranes = [∑(βα-22(*S* + *R*)-diastigmastane)]/[∑(ααα- and αββ-20(*S* + *R*)-stigmastane)]; ^4^ Ts = 18α-22,29,30-tris*norneo*hopane; Tm = 17α-22,29,30-trisnorhopane. Steranes and hopanes identified using the MRM M^+^ → 217 and M^+^ → 191 transitions, respectively. ^5^ Pristane and phytane integrated using total ion chromatogram. * Value omitted—coelution with abundant C_27_ steroid hydrocarbons was such that value was not able to be determined.

**Table 4 biology-11-01289-t004:** Average δ^13^C values (bulk residue and compound specific) for PE 52316 and PE 52336, given in per mil (‰).

	PE 52316		PE 52336	
	Fossil	Matrix	Fossil	Matrix
δ^13^C_org_	−23.6 (0.07)^3^	−23.8 (0.03)^2^	−23.9 (0.12)^3^	−23.7 (0.03)^3^
δ^13^C_17_	−28.4 (0.65 *)^3^	−30.0 (0.24)^3^	−29.4 (0.54 *)^3^	−29.0 (0.62 *)^3^
δ^13^C_Pristane_	−28.8 (0.82 *)^2^	−29.2 (0.30)^3^	−28.4 (0.53 *)^3^	−29.3 (0.51 *)^3^
δ^13^C_18_	−30.4 (0.41 *)^3^	−31.7 (0.31)^3^	−31.5 (0.13)^3^	−30.7 (0.40)^3^
δ^13^C_Phytane_	−33.0 (0.11)^2^	−29.5 (0.23)^3^	−35.1 (0.39)^3^	−33.1 (0.31)^3^
^1^δ^13^C_20–25_	−30.5	−31.0	−30.8	−30.7
δ^13^C_cholestane_	−32.9 (0.39)^3^	-	−32.6 (0.29)^3^	-

* Number in brackets indicates standard deviation; superscript refers to number of analyses used in average. Values with standard deviations greater than 0.4‰ are marked with an asterisk (*). ^1^δ^13^C values for C_20–25_
*n*-alkanes represent an average of the δ^13^C values for each *n*-alkane from C_20_ to C_25_, which were each determined from the average of three isotopic measurements. Standard deviations ranged from 0.02–0.60.

## Data Availability

Source data for ChemoSpace analyses (Figure 4 and Figure 5) are available in the Appendix A. All other data can be made available upon request from corresponding authors.

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
