# Peer review of "Fossil Biomarkers and Biosignatures Preserved in Coprolites Reveal Carnivorous Diets in the Carboniferous Mazon Creek Ecosystem"

_biology, 2022, doi:10.3390/biology11091289_

Round 1

Reviewer 1 Report

This is a well-conducted study utilising multiple analytical approaches to characterise Carboniferous coprolites. The team are adept at this type of study and they generate compelling evidence for a coprolite origin from carnivores. It is a shame that the authors haven’t included a chromatogram of the stenol/stanol analyses obtained for PE52315; it is an example of exceptional preservation. I would advise adding it as it really is the ‘smoking gun’ given the coelution of coprostane with C27 ααα 20S cholestane. The conclusions read rather tersely and could may benefit from a bit more contextualising and exposition about the real importance of these results to the wider field (to avoid any accusation of ‘stamp collecting’) and discussion of future directions. I have a few minor niggles below that should be addressed but otherwise am happy that this manuscript then be published.

Specific comments

1.   Line 65 – Change ‘campestanol’ and ‘stigmastanol’ to ‘5β-campestanol’ and ‘5β-stigmastanol’

2.   Line 68 – Change ‘in molecular’ to ‘from molecular’

3.   Line 72 – It is the ubiquity of cholesterol from which cholestane is derived rather than the ubiquity of cholestane in animal tissue which the authors state.

4.   Lines 88-128 – This section that sets up preservation and the biomarker concept should be moved to after line 58 so that it comes prior to discussion of any biomarkers, e.g. cholestane. Big picture à Biomarker concept à Your study

5.   Line 103 – Add ‘classes’ after ‘other’.

6.   Line 129 – Change ‘Accuracy and detail of the’ to ‘The accuracy and detail of’.

7.   Line 130 – ‘Biosignature’? Do you mean ‘biomarker’ as you have not defined this term.

8.   Line 152 – Stick with 34S/32/S, 13C/12C and 18O/16O as you do earlier and leave the delta notation for when you are talking about actual delta values.

9.   Line 179 – Change ‘mortar’ to ‘mortars’;

10.Lines 198-204 – Whilst the identities of all of the fragments represented by these m/z values are well-established you should still list them all in the SI or refer the reader to an appropriate reference.

11.Lines 200 and 203 – Delete ‘Daltons’, it is not needed (or correctly used here).

12.Lines 216-220 – This is incorrect. ‘MRM’ stands for ‘multiple reaction monitoring’ although even this was just a differentiator term invented by a manufacturer. Better to refer to the monitoring of transitions as ‘selected ion monitoring (SRM)’. Likewise, use ‘selected ion monitoring (SIM)’ rather than ‘selected ion recording (SIR)’. You correctly use SIM earlier so this will make things consistent.

13.Line 222 – It is a δ13C value determination derived from a stable isotope ratio mass spectrometric analysis. Correct here and anywhere else this error occurs.

14.Lines 224-225 – The reactorsare not composed of a combustion interface, that is where you put them! The ceramic tube contained a NiO tube with NiO and CuO wires.

15.Line 228 – Use GC-C-IRMS, i.e. include and define the reactor (in this case ‘combustion’).

16.Lines 229-230 – Change ‘measuring the abundances of ions of’ to ‘measuring the response of’. The Faraday cups are not measuring individual ions like in an organic MS.

17.Line 248 – ‘cholesterol’ not ‘Cholesterol’

18.Line 249 – ‘N’ and ‘O’ should be italicised and the ‘Bis’ changed to ‘bis’

19.Line 269 – Superscript ‘-1’

Line 357 – See earlier comment about MRM.

Author Response

Please see the attachment. We would like to sincerely thank the reviewer for their helpful and constructive comments.

Reviewer 2 Report

Your work is very interesting. This provides a novel approach for coprolites analyses, especially in samples of that antiquity. I suggest in future analyses to complement this chemical approachs with other multiproxy studies that to provide evidences of diet (e.g., pollen, plant remains, ancient DNA).

Author Response

We would like to sincerely thank the reviewer for their feedback, as well as their thoughtful considerations for future analysis.

Reviewer 3 Report

The manuscript describes molecular analysis of coprolites to obtain information about the creature that produced them. The implications of the work are large and the results are interesting. There are several assumptions and details that could be better justified to make the paper stronger, but generally the work is publishable after these minor revisions.

(1) The authors imply (in abstract) that something is endogenous if it is found in the coprolite and not in the surrounding sediment, but that doesn't seem to rule out action/remains of microbial or fungus, for example, since their action will be different where the nutrition is different (at least early in the process). The authors should be a little more careful with such claims, and provide further arguments, especially where there are other possibilities (e.g., line 391).

(2) The samples were obtained from a museum (line 70). What is the history and treatments to these samples? How can we be sure that they have not been contaminated? Some of the techniques used are only surface-sensitive.

(3) Another assumption (used in e.g., lines 85-87 or 131-133) that needs further justification is that these specific molecules will be preserved in sites with preservation of soft tissue patterns. Why does one imply the other?

(4) The Raman analysis does not stand on its own in this manuscript. Familiarity with the author's prior work or in-depth knowledge of ChemoSpace is required. More details should be provided here at least in summary form.

(5) The Raman spectrum processing also needs to be discussed. In particular, was any processing done besides spike removal? What background subtraction, if any, was used?

(6) In Fig. 8, please label 'Axis 1' and 'Axis 2' or explain them qualitatively. Why can we assume that digestion moves the measurements to the right in that plot (besides circular reasoning)?

Author Response

Please see the attachment. We would like to sincerely thank the reviewer for their constructive comments that have helped to improve the manuscript.
